# Uniaxial Compression Failure and Size Effect of Recycled Aggregate Concrete Based on Meso-Simulation Analysis

**DOI:** 10.3390/ma15165710

**Published:** 2022-08-18

**Authors:** Jingbo Zhuo, Yamin Zhang, Mei Ma, Yu Zhang, Yuanxun Zheng

**Affiliations:** 1Yellow River Laboratory, Zhengzhou University, Zhengzhou 450001, China; 2School of Water Conservancy Engineering, Zhengzhou University, Zhengzhou 450001, China; 3Library, Zhengzhou University, Zhengzhou 450001, China

**Keywords:** recycled aggregate concrete, interfacial transition zone, maximum aggregate size, size effect, meso-compression numerical simulation

## Abstract

Recycled aggregate concrete (RAC) is a kind of five-phase composite material at the meso-level. It has a more complex interfacial transition zone (ITZ) than ordinary aggregate concrete (NAC), which is an important factor affecting the meso-failure of RAC. In addition, the maximum aggregate size plays an important role in the nonlinear mechanical behavior of concrete, which is closely related to the size effect. In this paper, a 2D random aggregate model of RAC is established based on meso-mechanics. The mechanical properties and failure modes of RAC under uniaxial compression are simulated using a plastic damage constitutive model. Through variable parameter analysis, the effects of the properties and thickness of ITZ on the elastic modulus and peak stress of RAC are studied, and the effect of the maximum aggregate size on the size effect of the compressive strength of RAC is discussed. The results show that the ITZ strength has a positive linear correlation with the peak stress and elastic modulus of RAC, while the ITZ thickness has a negative linear correlation with the peak stress and elastic modulus of RAC. Under the same specimen size (D = 100 mm, 150 mm, 200 mm, 300 mm), with an increase in the maximum aggregate size (d_max_ =20 mm, 25 mm, 30 mm, 35 mm), the nominal compressive strength of RAC increases by 6–10%, and the size effect is gradually weakened. When the maximum aggregate size reaches 30 mm, a decrease in the size effect tends to slow down compared with the maximum aggregate size of 20 mm. The classical Bažant size effect law is applicable to describe the compressive properties of RAC under different maximum aggregate sizes, and has a certain guiding significance for the prediction of the size effect of RAC in practical engineering.

## 1. Introduction

Natural aggregate is widely used in concrete buildings, and its content accounts for about 60–70% of the total volume of concrete [1]. The extraction of natural aggregates consumes a large amount of natural resources and adversely affects the global environment [2,3,4]. In addition, waste concrete is one of the environmental problems that needs to be solved urgently at present [5]. Preparing recycled coarse aggregate (RCA) from waste concrete and applying it to concrete is one of the important measures to solve this problem, and can effectively solve the problem of natural aggregate shortages. However, the main difference between RCA and natural aggregate is that RCA has the characteristics of rough surface and low mechanical strength due to the adhesion of old cement mortar and micro-cracks in the process of re-crushing [6]. RCA can be regarded as a miniature concrete block with the complex structure of inhomogeneous composite materials, which limits the large-scale application of RCA in practical engineering. Li et al. [7] found that the micro-structure of RCA is different from that of natural aggregate, and there is a wall effect around RCA, which makes the water–cement ratio of ITZ relatively high, resulting in a large number of pores and cracks near it. In order to solve the problem of low-quality RCA, scholars modified RCA by means of physical strengthening [8], chemical strengthening [9], and microbial strengthening [10], which effectively improved the physical properties of RCA, such as the crushing value, water absorption, apparent density, and so on. Therefore, the study of the ITZ composition and strengthening methods of RCA is of great significance to the application and study of the meso-composition of RAC.

RAC is a five-phase complex structure composed of natural aggregate, adhered mortar, new cement mortar, and new and old ITZ [11]. Recycled coarse aggregate (RCA) extracted from abandoned buildings is considered to be a feasible and sustainable substitute for natural aggregate [12]. RAC is the concrete prepared by crushing waste concrete into coarse aggregate and replacing natural aggregate in whole or in part [13,14]. The use of RAC can not only reduce environmental pollution, but also reduce the extraction rate of natural aggregate, which is consistent with the concept of sustainable development [15,16]. However, the existence of adhered mortar of RCA makes the composition of ITZ in RAC more complex than that of ordinary aggregate concrete (NAC). The natural defects such as brittleness and easy cracking of RAC are more prominent, and the mechanical properties and durability of NAC are worse than those of NAC [17,18]. At present, scholars have carried out a large number of basic experimental studies and achieved a lot of results, but due to the difference in experimental conditions, the instability of RCA sources, and many other factors, the research conclusions are quite different, even contradictory [18,19]. At the same time, the development of microcracks inside specimens cannot be accurately observed in the field test process, and the research on meso-damage mechanism, multi-axial strength, and constitutive relation is insufficient [20,21]. To sum up, it is difficult to reveal the essence of mechanical behavior and failure mechanism of RAC from a single macroscopic point of view, while meso-numerical analysis can overcome the limitations of macroscopic mechanical tests of RAC, thus realizing a more in-depth study on the macroscopic mechanical properties, durability, and practicability of RAC.

In the past, concrete was regarded as a homogenous material in macroscopically isotropic in classical mechanics [22], but the internal structure of concrete was complex and highly different [23,24]. The generation, expansion, and penetration of concrete cracks are not only affected by its internal composition, but also by the spatial distribution, volume fraction, and gradation of aggregate. The theory of meso-mechanics assumes that concrete is a heterogeneous three-phase composite composed of aggregate, fresh mortar, and ITZ between them [25]. With the development of meso-mechanics, scholars have proposed many meso-mechanical models, such as the M-H model [26], the random particle model [27], the lattice model [28], and so on. Xiong et al. [29] established a 3D multiphase meso-scale model, which analyzed the evolution process of damage and transport properties of concrete, and promoted the relationship between meso-simulation and macro-scopic tests. In addition, both the LDPM model proposed by Cusatis et al. [30] and the RBSM model proposed by Nagai et al. [31] can accurately reflect the mesoscopic failure mechanism of concrete and have been well applied in reinforced concrete. However, RAC is a five-phase composite material with more complex micro-structures and mechanical properties than NAC, so it is more difficult to realize the RAC failure process through numerical simulation.

In the numerical simulation of RAC at the mesoscopic level, a heterogeneous composite model consisting of natural aggregate, new mortar, attached mortar, and old and new ITZs was first established. Due to the different mechanical properties of the mesoscopic components of RAC, the failure mechanism also changed. Based on the meso-structure model of concrete, the meso-structure model of RAC has also been rapidly developed. Xiao et al. [32] combined the lattice model with the random aggregate model to analyze the influence of various meso-components of RAC on its mechanical properties under uniaxial compression. Peng et al. [33] effectively verified the relationship between the micro-structure and mechanical properties of RAC using the BFEM method, and simulated the whole process of the compressive failure of RAC specimens of different sizes. Compared with NAC, Xiao et al. [34] and Tam et al. [35] found that the micro-structure of RAC was much more complex than that of ordinary concrete. In order to study the effects of ITZ on the meso-scopic mechanical properties of RAC, Jayasuriya et al. [36] discussed the effects of ITZs and the contents of adhered mortar on the mechanical properties of RAC through a bivariate contour map, and concluded that ITZ and the contents of adhered mortar are the main parameters causing a reduction in the strength of RAC. Etxeberria et al. [37] and Etxeberria et al. [38] also obtained similar conclusions. In addition, Xiao et al. [39] studied the stress distribution of RAC under uniaxial compression through the nine-particle aggregate model, and concluded that the ratio of old ITZ to adhered mortar has a significant impact on the stress–strain curve and failure process of RAC. However, the ITZ of RAC is a complex structure, and it has always been a challenge to evaluate the stiffness of its material properties [40,41]. In addition, the thickness of ITZ is also one of the main topics for studying the behavior of RAC, which is of great significance to the development of RAC [42,43]. In summary, based on the random aggregate model, on the premise of exploring the random distribution of aggregates and boundary conditions, the effects of new and old ITZ strength and thickness on the uniaxial compression failure and stress–strain relationship of RAC are studied quantitatively.

At present, concrete materials are widely used in super high-rise buildings, long-span bridges, and other large-size structures. However, with an increase in the structure size, the mechanical property index represented by the compressive strength of concrete is no longer a fixed value, but changes with the change in the geometric size of the material; thus, there is a size effect [44,45]. Currently, there are different research theories on the phenomenon of the size effect. Bažant et al. [46], Weibull et al. [47], and Carpinteri et al. [48] have proposed a variety of size effect formulas, and the rationality of these formulas is verified by the test results in a certain range. In addition, a large number of field tests and numerical simulation studies show that the size effect of concrete is mainly affected by the aggregate size, aggregate distribution, aggregate shape, initial defects, and other factors [49,50]. Among them, aggregate size is considered to be one of the most important factors affecting the mechanical properties and failure mechanism of concrete, and has an important influence on the mechanical behavior and size effect of concrete. Jin et al. [51] studied the effect of aggregate size (24–42 mm) on the cracks of internal components of concrete with different sizes (100–450 mm) during compressive failure. It is concluded that an increase in aggregate size can improve the sensitivity of strength to structural size. On the contrary, Jin et al. [52] found that an increase in aggregate size could reduce the sensitivity of the size effect when studying the size effect of different aggregate sizes on the splitting tensile strength of concrete. Compared with NAC, the field test shows that the size effect of mechanical properties of RAC is more significant. However, at the meso-scale, there are few studies on the size effect of RAC, and even fewer studies on the effect of different aggregate sizes on the compressive size effect of RAC. In addition, Bažant’s size effect law has rarely been used to verify the size effect of RAC. Therefore, it is of great research value to conduct a numerical simulation of RAC specimens with different particle sizes and structural sizes and to study the size effect of RAC compressive strength.

In summary, due to the complexity of the composition of ITZ, the compression failure behavior of RAC is more complex, the research conclusions are also diversified, and the understanding of its mechanisms is insufficient and disjointed. Moreover, compared with the size effect of NAC, the size effect of RAC is more complex. However, due to the limitations of testing equipment and conditions, it is difficult to carry out physical tests on the failure of larger RAC specimens to reveal the law of size effect. Therefore, based on the numerical method of mesomechanics, a 2D random aggregate model of RAC is established to simulate the stress–strain curve and failure mode of RAC under the uniaxial compression load. The effects of the strength and thickness of new and old ITZ on the elastic modulus, peak stress, and failure mode of RAC are obtained through variable parameter analysis. Based on the validation of the numerical model, the influence of maximum aggregate size on the failure modes and stress–strain curves of RAC with different sizes is investigated, and the applicability of Bažant’s size effect law to RAC is verified.

## 2. Establishment and Verification of the Meso-Model of RAC

### 2.1. Establishment of the Meso-Model of RAC

RAC is a complex five-phase material at the meso-level, where the aggregate acts as a support in the overall concrete structure. RAC and NAC have similar influencing factors, and their physical and mechanical properties are affected by the gradation, randomness, and delivery rate of RCA. In order to conform to the composition of the real RAC and to meet the computational efficiency and accuracy of the solution, a reasonable and simplified meso-scopic finite element model is used to analyze the RAC. The establishment of the meso-model is mainly to generate aggregate particles with random distribution. In order to characterize the inhomogeneity of the RAC meso-scale in detail, polygonal aggregates are used to establish the meso-model of RAC in this section. The specific modeling process is as follows.

**Step 1.** The random RCA structure is generated based on the Monte Carlo method, and the randomness of meso-structure is transformed into mathematical random variables, which conform to uniform distribution. This section mainly introduces the 2-D random aggregate generation criterion; generates pseudo-random numbers in a given interval with the help of Python language; generates aggregate coordinates, interfaces, and other information; and completes the steps of the generating model, assigning attributes, meshing, post-processing, and so on in ABAQUS.

**Step 2.** Based on the Fuller grading curve [53] and the Walraven formula [54], the aggregate distribution and quantity of each particle size are determined. The Fuller gradation curve is combined into the continuous gradation ratio of the minimum void according to different proportions to make the concrete achieve the maximum compactness. The theoretical formula is as follows.
(1)$$P=100\sqrt{\frac{D_{0}}{D_{max}}}$$
where *D*_0_is the diameter of the sieve hole, *P* is the mass percentage of aggregate passing through the sieve hole *D*_0_, and *D_max_* is the maximum aggregate particle size.

The Fuller gradation curve is suitable for the 3D model. In the 2D aggregate model, the Walraven gradation formula is generally used to transform the 3D gradation curve to the 2D state. The theoretical formula is as follows.
(2)$$\begin{array}{ll} P_{C}\left(D<D_{0}\right)= & P_{k}(1.065{\left(\frac{D_{0}}{D_{max}}\right)}^{0.5}-0.053{\left(\frac{D_{0}}{D_{max}}\right)}^{4}-0.012{\left(\frac{D_{0}}{D_{max}}\right)}^{6}\\ & -0.0045{\left(\frac{D_{0}}{D_{max}}\right)}^{8}-0.0025{\left(\frac{D_{0}}{D_{max}}\right)}^{10}) \end{array}$$
where $P_{k}$ is the ratio of the aggregate volume to the total volume of specimen, and the others are the same as Formula (1) above.

Based on Formulas (1) and (2), this paper adopts primary RAC. The particle size range of RCA is 5~20 mm, which is divided into three sizes: 5~10 mm, 10~15 mm, and 15~20 mm, and the aggregate volume fraction of each size is the same.

**Step 3.** There are many ways to generate random polygon aggregate. This paper adopts the polygon characteristic parameter transformation method, as shown in Figure 1. The specific generation process is as follows. (1) Regular polygons with random centroid and random radius are generated according to gradation. The centroid of a regular polygon is connected to each vertex, and the initial angle $\alpha_{i}\left(i\leq n\right)$ of every two adjacent segments is 2π/n, where n is the number of sides of the polygon. (2) The characteristic parameters of regular polygons are randomly transformed by transform coefficients. Among them, each $\alpha_{i}$ and another randomly selected feature angle $\alpha_{j}$ are randomized according to the following formula.
(3)$$\left\{\begin{array}{c} \alpha {’}_{i}=\alpha_{i}+\gamma \times min\left(\alpha_{i},\alpha_{j}\right)\\ \alpha {’}_{j}=\alpha_{j}+\gamma \times min\left(\alpha_{i},\alpha_{j}\right) \end{array}\right.$$
where γ is the angular transformation coefficient, and $\alpha {’}_{i}$ and $\alpha {’}_{j}$ are the randomized characteristic angles. The characteristic radius $r_{k}$ of the regular polygons is randomized, according to the following formula.
(4)$$r{’}_{k}=r_{k}+rand\left(-r_{k}\times \omega ,r_{k}\times \omega \right)$$
where ω is the radius transformation coefficient, and $r{’}_{k}$ is the characteristic radius after randomization.

(3) The vertex coordinate $\left(x_{i},y_{i}\right)$ of polygonal aggregate is calculated according to the characteristic angle and characteristic radius after randomization, and it is decided whether the vertex coordinate is in the specimen area. If all vertices are inside the area, the aggregate is kept, or otherwise discarded, and the new aggregate is generated in the same way until the requirements are met.

**Step 4.** The establishment of the meso-finite element model. Based on the study of aggregate shape in the meso-simulation by Sadouki et al. [55] and Cusatis et al. [30] and the shape of RCA in the actual engineering, three aggregate particle sizes of 7, 12, and 20 mm are selected within the particle size range of 5–20 mm for convenient calculation. The area fraction of RCA is about 45%, and the area fraction of the three particle sizes is 15%. The classical “take-place” method is used to randomly place aggregate [56,57], and the random aggregate model is generated, as shown in Figure 2, and the phases of the random aggregate model are shown in Figure 3.

### 2.2. Constitutive Model

The fundamental difference between RAC and NAC is that the coarse aggregate is surrounded by a layer of cement mortar, and the adhered mortar can be regarded as a weakening mortar, so it is reasonable to use the classic plastic damage model of NAC in the numerical simulation of RAC. The strength of NCA is higher than that of mortar matrix and ITZ, so there is no penetrating crack under compression. Therefore, NCA is defined as a linear elastomer. The mechanical behavior of cement mortar and ITZ is similar to that of concrete, and the mechanical behavior of concrete is described by the CDP model of concrete in ABAQUS. The CDP model was established on the basis of Lubliner [23] and Lee [24] models, and the following simplified assumptions were made: (1) the continuity of concrete; (2) the isotropy of injury; and (3) the determination of the failure surface with two parameters (the equivalent plastic compressive strain ${\tilde{\varepsilon \;}}_{c}^{\;pl}$ and the equivalent plastic tensile strain ${\tilde{\varepsilon \;}}_{t}^{\;pl}$).

The CDP model considers the crack propagation, damage, and stiffness degradation of concrete under cyclic loading, and assumes that the two main failure mechanisms of concrete are tensile cracking and compression crushing. Under uniaxial tension, the stress–strain curve is elastic before the peak stress and softens after exceeding the peak stress. In uniaxial compression, linear elasticity occurs before the stress reaches the initial stress, hardening occurs between the yield stress and the ultimate stress, and softening occurs after the ultimate stress. The stress–strain curve of the CDP model under tension and compression is shown in Figure 4, and the stress–strain relationship can be expressed as follows.
(5)$$\sigma_{t}=\left(1-d_{t}\right)E_{0}\left(\varepsilon_{t}-{\tilde{\varepsilon \;}}_{t}^{\;pl}\right)$$
(6)$$\sigma_{c}=\left(1-d_{c}\right)E_{0}\left(\varepsilon_{c}-{\tilde{\varepsilon \;}}_{c}^{\;pl}\right)$$
where $\sigma_{t}$ and $\sigma_{c}$ are the tensile stress and the compressive stress, respectively; $\varepsilon_{t}$ and $\varepsilon_{c}$ are the tensile and compressive strains, respectively; $E_{0}$ is the initial elastic modulus; ${\tilde{\varepsilon \;}}_{t}^{\;pl}$ and ${\tilde{\varepsilon \;}}_{c}^{\;pl}$ are plastic tensile strain and plastic compressive strain, respectively; and $d_{t}$ and $d_{c}$ are the tensile and compressive damage factors, respectively (0 means no damage, 1 means complete damage).

The softening stage in the stress–strain curve of concrete is defined by setting the inelastic strain ${\tilde{\varepsilon \;}}_{c}^{\;in}$ under compression and the strain $\varepsilon_{t}^{\;ck}$ under tensile cracking. In ABAQUS, the inelastic strain ${\tilde{\varepsilon \;}}_{c}^{\;in}$ under compression and the strain $\varepsilon_{t}^{\;ck}$ under tensile crack are transformed into an equivalent plastic compressive strain $\varepsilon_{c}^{\;pl}$ and an equivalent plastic tensile strain $\varepsilon_{t}^{\;pl}$ using Formulas (7) and (8).
(7)$$\varepsilon_{c}^{\;pl}={\tilde{\varepsilon \;}}_{c}^{\;in}-\frac{d_{c}}{\left(1-d_{c}\right)}\frac{\sigma_{c}}{E_{0}}$$
(8)$$\varepsilon_{t}^{\;pl}={\tilde{\varepsilon \;}}_{t}^{\;ck}-\frac{d_{t}}{\left(1-d_{t}\right)}\frac{\sigma_{t}}{E_{0}}$$

The results show that the tensile strength of the mortar matrix is generally 1/10 of the compressive strength [33,36,58,59]. Generally, ITZ is regarded as porous and weakened mortar matrix. In particular, the mechanical parameters of ITZ (elastic modulus, compressive strength, and tensile strength) are reduced to different degrees as mechanical parameters of mortar matrix, and numerical simulation is carried out many times until they are in good agreement with the test results.

### 2.3. Verification of the Meso-Model

In order to discuss the compressive performance of RAC, the strength design grade of C30 is used for pouring RAC in this paper, and the mixing ratio is designed according to “General Portland cement” (China GB175-2007). As shown in Table 1, the water–cement ratio is 0.5, the weight percentage of fine aggregate to total aggregate is 36%, the RCA replacement rate is 100%, and the particle size of RCA is 5–20 mm. The RCA, purchased from Xuchang Jinke Resources Recycling Co., Ltd., is prepared from crushed waste concrete. The crushed waste concrete is screened by two screens with a maximum pore size of 20 mm and a minimum pore size of 4.75 mm, and the fragments between 5 mm–20 mm are taken as RCA. The particle size distribution of RCA is shown in Figure 5. The specimen size of compressive strength is 150 mm × 150 mm × 150 mm. The RAC and mortar specimens with the age of 28 days are measured. The test results of 28-day compressive strength of RAC and mortar are shown in Table 2. The standard deviation in the table is expressed by the symbol σ, and the coefficient of variation is expressed by CV. In order to verify the correctness of the model, the simulation results are compared with the results of RAC uniaxial compression test carried out in the field. As shown in Table 3, the strength proportional relationship between the components of RAC is given according to the existing research, including new ITZ and mortar [59,60,61], new and old ITZ [62,63], and new and old mortar [64,65,66]. The strength of mortar is defined as 1 in Table 3. The compressive strength and elastic modulus of mortar are obtained from the test, and the parameters are substituted into the model. The specific parameters are shown in Table 4.

Three groups of RAC meso-models with a size of 150 × 150 mm are established, in which the content of coarse aggregate is 45% and the replacement rate is 100%. The simulated stress–strain curves and failure patterns are compared with the field test data, as shown in Figure 6. The results show that the stress–strain curves obtained by numerical simulation are in good agreement with those obtained by the test. The ascending part of the curves is approximately the same, while the descending part is slightly different. The errors of peak stress and peak strain are within 5%. Figure 7 shows the final failure result of the model. It can be seen that the final failure state of the model presents a cone-angle failure mode with multiple inclined cracks penetrating through the specimen, and the final cracking path is similar to the failure mode obtained in the test. In conclusion, the applicability of the meso-mechanical method to RAC and the rationality of material parameter selection are verified. When meso-numerical simulation is carried out by ABAQUS, the damage factor corresponding to the color bar can reflect the damage degree of the simulated sample. When the damage factor is 0, there is no crack in RAC, and the loading does not begin, indicating that the sample is not damaged. With an increase in the displacement load, the damage factor changes from 0 to 1. At first, the specimen forms a micro-crack from the ITZ, propagates along the edge of the aggregate, and spread to the mortar. Then, the damage spreads all over the stress area of the sample, and finally forms the macroscopic crack. At this time, the sample is damaged. When the damage parameter is greater than 0.9, it is considered that the cracks occur in concrete macro-scopically. When the damage factor reaches 1, the displacement load reaches the maximum value, and the sample is completely damaged. Yu et al. [67], Chen et al. [68], Benkemoun et al. [69], Li et al. [70], and Jin et al. [71,72] adopted similar expressions for color bars in previous meso-numerical simulations of concrete.

### 2.4. Analysis of the Boundary Conditions and Random Distribution of RAC

The results of several studies show that when uniaxial compression numerical simulation is carried out, failure modes with an “X” shape [68,73] and a “V” shape [74,75] as the main failure modes will be formed when different constraints are applied to the bottom of the specimen. As shown in Figure 8a, transverse restrictions on the bottom and top of the microscopic specimen are ignored, and fixed hinges are used to support the midpoint of the bottom. As shown in Figure 8b, fixed hinges are used to support each point at the bottom of the specimen. Under the two kinds of constraint conditions, the failure mode of the samples of RAC occurs under high friction and low friction, respectively. As shown in Figure 9, inclined cracks appear in RAC under high friction constraints, and the failure shape is similar to “V”, while the two-way failure mode appears when low friction constraints are involved, and the failure shape is similar to “X”. It can be seen from Figure 6 that the compression test block in the field test is mainly V-shaped failure, so the boundary condition 1 is selected for meso-numerical simulation under various working conditions.

In the actual construction process, RCA is randomly distributed in concrete due to vibration and mixing, which affects the overall performance of RAC. Therefore, the study of random distribution of RCA is of great significance to the mechanical behavior of uniaxial compression. In this section, five groups of RAC numerical models with different random distribution states are established to study the influence of random distribution of RCA on the overall macroscopic mechanical properties. Figure 10 shows the stress–strain curve of RCA randomly distributed under uniaxial compression. The results show that the stress–strain curves are almost the same before the peak stress, but the softening part of the curves is slightly different. This is mainly because the number of cracks increases and develops further after the peak stress, and the different distribution of RCA will change the path of crack propagation, thus affecting the descending section of the stress–strain curve.

Figure 11 shows the failure patterns of RAC under a different random distribution of RCA, and the damage distribution is a typical V-shaped failure mode. The micro-crack at the interface further develops and evolves under the external load, from the interface layer to the mortar unit, causing damage to the mortar unit, and finally forming through cracks along the direction of the maximum shear stress.

In summary, when considering the influence of other factors on the stress–strain curve of RAC, the error caused by the random distribution of RCA can be ignored.

### 2.5. Study on Mesh-Dependence

In the meso-modeling of RAC, the irregular geometry of random aggregate and ITZ can easily lead to poor mesh quality and unreasonable numerical results. In order to study the influence of mesh size on meso-scopic simulation results, different mesh sizes are compared and analyzed. The minimum size of aggregate established in this paper is 5 mm, and the minimum unit area should be less than 1/2 of the minimum particle size of aggregate in microscopic numerical simulation. Therefore, meshes with three sizes of 0.5 mm, 1 mm, and 1.5 mm are selected for analysis in this section, and the mesh shapes divided in this paper are mainly quadrilateral. An eight-thread computer with a 2.3 GHz CPU is used for calculation. The number of elements and computing times of the meso-scopic model of RAC under different mesh sizes are shown in Table 5.

Figure 12 shows the stress–strain curves of RAC under different sizes of mesh. Although the results are slightly different, the ascending section, peak stress, and softening section are very close, indicating that the stress–strain behavior of RAC is mesh-insensitive. The failure modes of RAC of the three sizes of mesh are shown in Figure 13. It can be seen that the size of mesh has no significant effect on the general uniaxial compression damage path, and the smaller size of mesh shows a finer expansion path for local damage.

To sum up, considering the cost of calculation time, this paper sets the size of mesh of each component of RAC as 1 mm, and sets their element shape as quadrilateral.

## 3. The Effect of ITZ on the Compressive Properties of RAC

### 3.1. The Effect of the Strength of ITZ on the Compressive Properties of RAC

According to the weakest loop theory, ITZ is the weakest phase in RAC five-phase materials, and its strength determines the compressive strength of the specimen. ITZ is a complex structure, and it has always been a challenge to evaluate the stiffness of its material properties. In order to measure the strength of the ITZ and the relationship between the strength of the new and old ITZ and the strength of the new and old mortar, a large number of RAC specimens have been tested using nano-indentation [64,65,66]. Based on a large number of experimental data [62,63], the performance of the old ITZ is about 1.1 times that of the new ITZ, and the strength of the adhered mortar is about 1.1 times that of the new mortar. According to existing studies, the ratio of mechanical parameters of new ITZ to new mortar is generally 0.4~0.9 [59,60,61]. In view of this, the mechanical parameter ratios of the new ITZ and the new mortar are established as 0.4, 0.5, 0.6, 0.7, 0.8, and 0.9, respectively, to study the influence of the mechanical properties of ITZ on the stress–strain curve and failure mode of RAC.

The effect of ITZ properties on the stress and strain of RAC under uniaxial compression is shown in Figure 14a. As the strength and elastic modulus of ITZ are calculated as 0.4–0.9 of mortar, the peak stresses of RAC are 22.11 MPa, 23.43 MPa, 24.76 MPa, 25.70 MPa, 26.67 MPa, and 29.06 MPa, respectively. As shown in Figure 14b,c, the compressive strength and elastic modulus of RAC increase with an increase in the ratio of ITZ to mortar. Moreover, the increasing trend of compressive strength and elastic modulus is linear, with an increase in the ratio of ITZ to mortar ranging from 0.4 to 0.9, and the strength of RAC is increased by 19.50% and 23.92%, respectively.

Figure 15 shows three groups of RAC compression failure modes with different strengths of ITZ. It can be observed that the final compression damage mode of RAC does not change significantly with an increase in the strength of ITZ. The reason is that the compression damage first forms micro-cracks from ITZ, then spreads along the edge of aggregate, expands to mortar, and finally forms macro-cracks. As the RCA in the model is fixed in position, its crack propagation path is consistent, so the failure mode is similar. The difference is that the improvement in the mechanical properties of ITZ reduces the influence of ITZ on the bearing capacity of specimens to a certain extent, which strengthens the ability of ITZ to resist external loads, thus improving the strength of RAC.

To sum up, when considering the influence of other factors on the compressive properties of RAC, this paper chooses the new ITZ strength as 0.7 of the mortar strength for meso-numerical analysis, which is most suitable for the failure condition of RAC under the real condition.

### 3.2. The Effect of ITZ Thickness on the Compressive Properties of RAC

At present, the thickness of ITZ is also one of the main topics in the study of concrete performance. RAC has multiple complex structures of ITZ compared with NAC [42,43], and it is of great significance to study the thickness of ITZ for the development of RAC. Based on microscopic imaging technology, the thickness of ITZ is within the range of 10~50 μm [39]. However, since the thickness of ITZ is not significant compared to the diameter of RCA, ITZ with true thickness will cause numerical difficulties and increase computational costs in mesoscopic analysis. Therefore, the thickness of ITZ is usually set between 0. 1 mm and 1 mm for numerical calculation. In this study, the ITZ around polygon aggregate is set as the same thickness area, and the old and new ITZ are considered as the same thickness, and the thickness of the old mortar remains unchanged at 0.5 mm. Considering the actual ITZ thickness and numerical calculation efficiency, the ITZ thickness is set as 0.02 mm, 0.05 mm, 0.3 mm, 0.7 mm, and 1 mm, respectively, to study the influence of the thickness of ITZ on the stress–strain curve and failure mode of RAC uniaxial compressive performance.

It can be seen from Figure 16a and Table 6 that the compressive strength and elastic modulus of RAC decrease with an increase in the thickness of ITZ. In addition, the descending part of the curve becomes steeper and the residual stress decreases gradually with an increase in the thickness of ITZ. As shown in Figure 16b,c, with an increase in the thickness of ITZ, the fitting results of the thickness of ITZ with elastic modulus and peak stress show a linear trend, and the correlation coefficients are 0.9856 and 0.9569, respectively. This is because the thickness of ITZ increases, the weak area increases, and the mortar area decreases relatively; thus, the effective area that can resist external forces under external load gradually decreases. The crack propagation becomes easier, and the overall bearing capacity of RAC decreases, which makes RAC enter the softening section ahead of time and the brittleness is enhanced.

The damage mode of uniaxial compression of RAC caused by different thickness of ITZ is shown in Figure 17. The results show that with an increase in the thickness of ITZ, the damage degree of RAC increases, the number of cracks and main cracks increases, and the failure area increases significantly. This phenomenon occurs because when the thickness of ITZ is larger, the crack is easier to form in ITZ, and the internal crack extending from ITZ to mortar can be easily formed. Therefore, the number of cracks formed is higher, the crack development is more abundant, and the RAC is easy to peel off.

To sum up, when considering the influence of other factors on the stress–strain curve of RAC, this paper chooses the thickness of new and old ITZ as 0.3 mm for meso-numerical analysis, which is most suitable for the failure condition of RAC under uniaxial compression.

## 4. The Size Effect of RAC under Uniaxial Compression under Different Maximum Aggregate Sizes

Based on the mesoscopic numerical model established above, the compressive properties of RAC with different sizes and different maximum aggregate sizes are studied to reveal the influence of the maximum aggregate size on the size effect of RAC.

### 4.1. Compression Failure Mode of RAC under Different Maximum Aggregate Sizes

Figure 18 shows four groups of failure modes of RAC with different sizes. It can be seen that the failure modes of RAC with different sizes are similar. The angle between the crack zone of RAC and the boundary of the specimen is about 40°~60°, and obvious shear failure surface appears. With an increase in the size, the cracking mode of RAC changes from a single crack to a crack zone composed of multiple cracks. This is because the maximum aggregate size decreases relatively with an increase in the specimen size, and the blocking ability of aggregate to cracks is relatively weakened, thus forming crack zones with an increased number of complex forms. In addition, with an increase in the size of RAC, the area of ITZ increases relatively, and the energy accumulated in the specimen increases. When RAC is destroyed, the number of internal cracks increases significantly to dissipate more energy.

### 4.2. The Stress–Strain Relationship of RAC under Different Maximum Aggregate Sizes

The compressive stress–strain curves of RAC specimens with different sizes are shown in Figure 19a–d. It can be seen that under the same size of RCA, the peak strength and peak strain of RAC decrease with an increase in the specimen size, showing an obvious size effect. However, the slope of the rising section of the curve has little difference; in other words, it has little effect on the elastic modulus of RAC.

### 4.3. The Effect of the Maximum Aggregate Size on the Peak Compressive Strength of RAC

Figure 20a shows the effect of the maximum aggregate size on the peak compressive strength of RAC. It can be seen that the compressive strength increases gradually with an increase in the maximum aggregate size under the condition of the same size of RAC. This is because the ability of aggregate to prevent micro-cracks becomes stronger with an increase in the maximum particle size of aggregate. As a result, the bending degree of the crack increases, the roughness of the failure surface increases, the dissipated energy increases, and the compressive strength of RAC is improved. Figure 20b shows the variation of peak compressive strength of RAC with different sizes. It can be found that the compressive strength of RAC decreases with an increase in size, which has an obvious size effect. However, at the same size, the compressive strength of RAC decreases with a decrease in the maximum aggregate size. The smaller the maximum aggregate size is, the faster the compressive strength decreases with the RAC size.

Figure 21 shows the relationship between the compressive strength of RAC and the specimen size under different maximum aggregate sizes obtained by numerical simulation. The linear fitting slope $k=\Delta f_{t}/\Delta D$ is used to characterize the degradation of compressive strength. It can be seen that the compressive strength of RAC decreases with an increase in the specimen size, but the value of k is different. With an increase in the maximum aggregate size, the absolute value of the degradation slope decreases gradually, indicating that an increase in the maximum aggregate size (within the scope of this study) can reduce the sensitivity of the compressive strength to the size effect, thus weakening the size effect.

## 5. Analysis of the Size Effect of RAC under Uniaxial Compression Based on the Maximum Aggregate Size

### 5.1. Analysis of Size Effect Degree of RAC

In order to quantitatively describe the effect of the maximum aggregate size on the size effect of the compressive strength of RAC, the size effect degree is introduced. With the side length of 100 mm as the reference size, the size effect degree γ can be calculated as follows.
(9)$$\gamma_{b}=\frac{f_{cu,100}-f_{cu,b}}{f_{cu,100}}\times 100\%$$
where *b* is the side length, $f_{cu,b}$ is the compressive strength of the RAC when the side length is *b*, and $\gamma_{b}$ is the size effect degree of the RAC when the side length is *b*.

According to the definition of the size effect, the larger the size effect degree, the more obvious the size effect degree. The size effect degree of RAC of different maximum aggregate sizes can be obtained from Equation (9), as shown in Figure 22. When the maximum aggregate size is 20 mm, $\gamma_{150}$ is 1.1, 1.18, and 1.23 times of 25 mm, 30 mm, and 35 mm; $\gamma_{200}$ is 1.09, 1.17, and 1.22 times of 25 mm, 30 mm, and 35 mm; and $\gamma_{300}$ is 1.07, 1.15, and 1.2 times of 25 mm, 30 mm, and 35 mm, respectively. It can be seen that when the maximum aggregate size is 20 mm, the size effect degree is the largest, and thus the size effect is the most obvious. With an increase in the maximum aggregate size, the size effect degree decreases gradually, and it is the smallest when the maximum aggregate size is 35 mm. When the maximum aggregate size is about 30 mm, the decreasing trend of compressive strength becomes slower.

### 5.2. Theoretical Verification of the Size Effect of Bažant

In terms of the size effect theory of concrete, various size effect theories have been proposed [48,76,77]. Among them, Bažant, based on the theory of fracture mechanics and a large number of experimental data, has proposed a unified expression for the nominal strength of concrete that is consistent with the elastic–plastic theory.
(10)$$\sigma_{Nu}=\frac{Bf_{c}^{’}}{\sqrt{1+D/D_{0}}}$$
where $\sigma_{Nu}$ is the nominal strength of concrete, $f_{c}^{’}$ is the compressive strength of concrete, *D* is the size of the specimen, and *B* and *D*_0_ are the structure-dependent constants (obtained by fitting simulated data). In order to obtain *B* and *D*_0_, Equation (10) is transformed into a linear equation.
(11)$${\left(\frac{f_{c}^{’}}{\sigma_{Nu}}\right)}^{2}=\frac{1}{D_{0}B^{2}}D+\frac{1}{B^{2}}$$

Equation (11) can be transformed into a linear equation.
(12)y=Ax+C
where $y={\left(\frac{f_{c}^{’}}{\sigma_{Nu}}\right)}^{2}$, x=D, $A=1/D_{0}B^{2}$, *and*
$C=1/B^{2}$. $f_{c}^{’}$ is the compressive strength when the side length of RAC is 100 mm, and when the maximum aggregate particle size is 20 mm, 25 mm, 30 mm, and 35 mm, $f_{c}^{’}$ values are 27.28 mpa, 27.77 mpa, 28.43 mpa, and 29.12 mpa, respectively. Parameters A and C can be obtained through linear regression analysis, and the linear equation is obtained, as shown in Figure 23. Based on the above numerical results, the theoretical parameters of the size effect of Bažant on RAC at different maximum aggregate sizes are shown in Table 7.

As shown in Figure 24, the simulated data are fitted to the Bažant size effect law, where the horizontal line is the strength criterion and the line with a slope of −1/2 is linear to elastic fracture mechanics (LEFM). It can be seen that the simulated data are distributed around the curve of the Bažant size effect law; in other words, the Bažant size effect law can effectively describe the compressive strength size effect of RAC. With an increase in the size of the RAC specimen, the simulation data are closer to the straight line of linear elastic fracture mechanics, and thus the brittleness of RAC is greater during failure. On the contrary, the smaller the size of the specimen, the closer the simulation result is to the plastic strength line.

### 5.3. Application of Meso-Numerical Simulation in Practical Engineering

Due to the great difference in the composition of concrete materials, there is a significant difference between the experimental results and the actual fully graded concrete properties, which is difficult to reflect the real mechanical properties of concrete, the lack of science, and the rationality. Therefore, it is urgent to study the static and dynamic mechanical properties of mass concrete. In addition, with the development of large-scale engineering structure, all kinds of special-shaped and complex large-size reinforced concrete members are used more and more. The decision around whether the theory and method of reinforced concrete structure design based on the test results of small specimens are still suitable for large-size members has attracted more and more attention from academia and engineering circles.

Due to the limitations of test conditions and economy, it is very difficult to carry out large-scale mechanical properties of large-scale reinforced concrete members, and numerical simulation has become a promising alternative. In view of the particularity of concrete materials, the study of macroscopic mechanical properties of concrete from the mesoscopic point of view has become a starting point in the research field of concrete materials and components. It also forms the basis for the study of nonlinear mechanical behavior and size effect of mass concrete structures and large-size reinforced concrete members. The following are the outstanding applications of mesoscopic numerical simulation in the field of concrete.

(1)The meso-mechanical analysis model of concrete is extended to the three-dimensional solid model, which can effectively predict the whole process of damage to fracture of concrete members under external load. At present, it is widely used in the long-term health monitoring of bridges and dams.(2)Through mesoscopic numerical simulation, the influence trend of fiber and other admixture materials on the mechanical properties and durability of concrete members can be accurately reflected, which has important guiding significance for the construction of practical projects. For example, the ultimate load of steel fiber reinforced concrete for bridge expansion joints and basalt fiber-reinforced concrete for structural toughening should be verified.(3)The meso-numerical analysis of concrete can be combined with the advanced 3D printing technology, which can effectively simulate the strength of 3D-printed concrete and other information. For practical projects, it can save a lot of manpower and material resources.(4)Meso-numerical analysis is the research basis of large-scale building components, such as seismic analysis of nuclear power plant and safety analysis of subway support design. Meso-analysis is the theoretical basis for the actual construction of mass concrete members.

In this study, the mechanical properties and compressive size effects of RAC are studied using mesoscopic numerical simulation. It can not only reduce various uncertain factors in field tests, but also predict the strength of large-size members of RAC, which is of far-reaching significance to practical engineering applications.

## 6. Conclusions

In this paper, the influence of the strength and thickness of ITZ on the compressive properties of RAC was studied, and the influence of the maximum aggregate size on the compressive strength size effect of RAC was discussed. The main conclusions are as follows.

(1)With an increase in the ITZ strength, the compressive strength and elastic modulus of RAC linearly increase with the slope of 10.85 and 12.82, respectively, and the failure mode of RAC under uniaxial compression basically remains unchanged. With an increase in the ITZ thickness, the compressive strength and elastic modulus of RAC decrease linearly with the slope of −5.71 and −4.38, respectively, and the failure mode of RAC under uniaxial compression changes from a single macroscopic main crack to multiple macroscopic cracks.(2)The compressive strength of RAC has a size effect under different maximum aggregate sizes. The larger the specimen size, the greater the brittleness, and the size effect is more obvious. With an increase in the maximum aggregate size, the compressive strength of RAC with a size of 300 mm decreases by 19.06%, 17.72%, 16.78%, and 15.76%, respectively.(3)At the same size, with an increase in the maximum aggregate size, the curvature of the crack and the roughness of the failure surface of RAC increase; thus, the compressive strength increases. With an increase in the size, the compressive strength increases by 6% and 10%.(4)Increasing the maximum aggregate size (within the range of this study) can reduce the sensitivity of the compressive strength of RAC to size, thus weakening the size effect. Among them, when the maximum aggregate size reaches 30 mm, a decrease in the size effect degree tends to slow down compared with the maximum aggregate size of 20 mm.(5)The classical Bažant size effect law can accurately describe the simulation results under different maximum aggregate sizes, which is suitable for the size effect analysis on the compressive strength of RAC. In addition, it has a certain guiding significance for the prediction of the size effect of RAC in practical engineering.

Although the data points at the maximum aggregate size are in good agreement with the size effect law of Bažant, there is a lack of quantitative correlation between an increase in the maximum aggregate size of the compressive strength of RAC and a decrease in the size effect. Establishing a size effect formula to quantitatively predict the nominal strength of RAC specimens of different sizes under different maximum aggregate sizes is a topic worthy of further discussion, and the author will carry out more in-depth theoretical research on this in the future. In addition, the meso-numerical study of RAC should be carried out in the following aspects in the future:(1)Study on the meso-mechanical behavior of RAC under three-dimensional load;(2)Meso-numerical study on the axial tensile and fracture properties of RAC;(3)Study on the dynamic failure of RAC under different strain rates;(4)Meso-numerical study on the failure of RAC using a phase field method.

## Figures and Tables

**Figure 1 materials-15-05710-f001:**
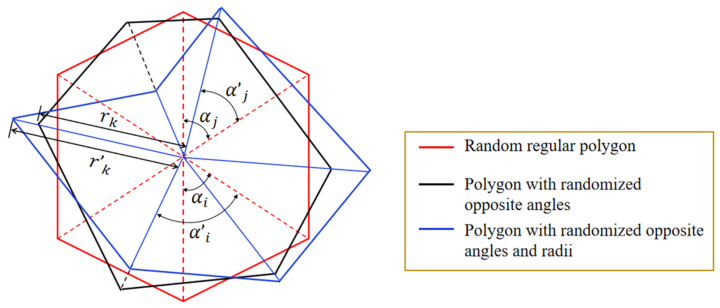
The generation process of random polygonal aggregate.

**Figure 2 materials-15-05710-f002:**
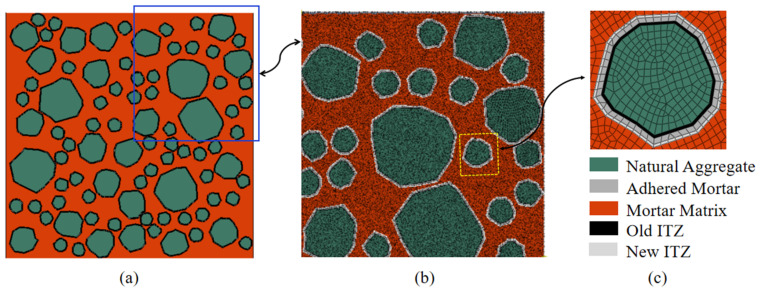
The meso-scopic model of RAC. (**a**) Aggregate model; (**b**) Meso meshing; (**c**) Meso componet.

**Figure 3 materials-15-05710-f003:**
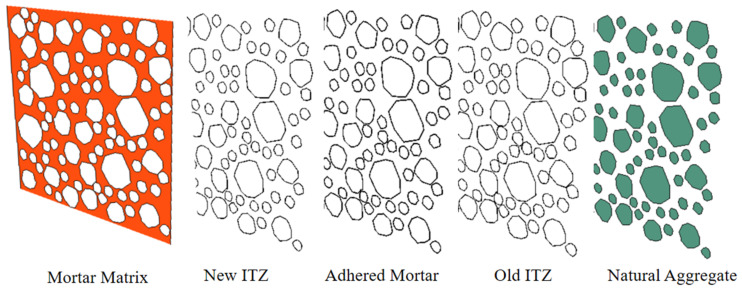
The element components for a 2D meso-model of RAC specimen.

**Figure 4 materials-15-05710-f004:**
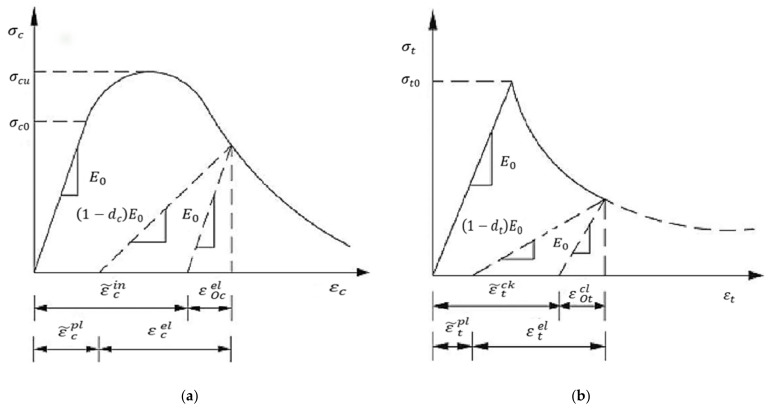
The CDP model. (**a**) Compression constitutive; (**b**) tensile constitutive.

**Figure 5 materials-15-05710-f005:**
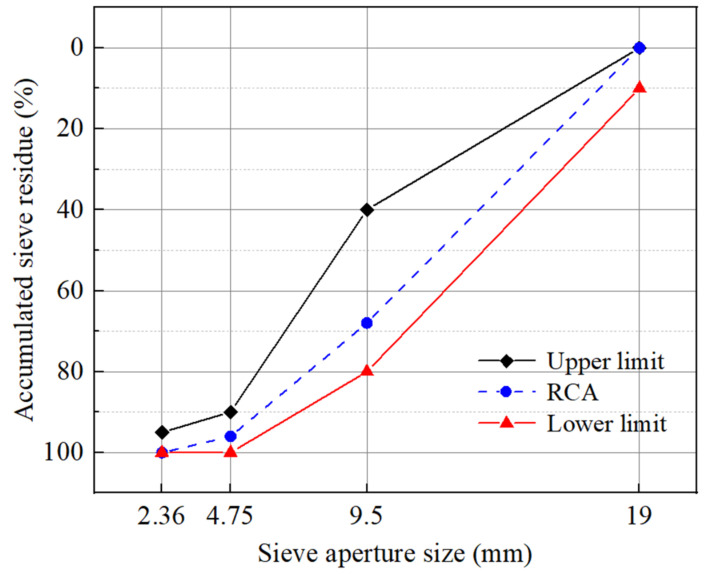
Gradation curve of RCA.

**Figure 6 materials-15-05710-f006:**
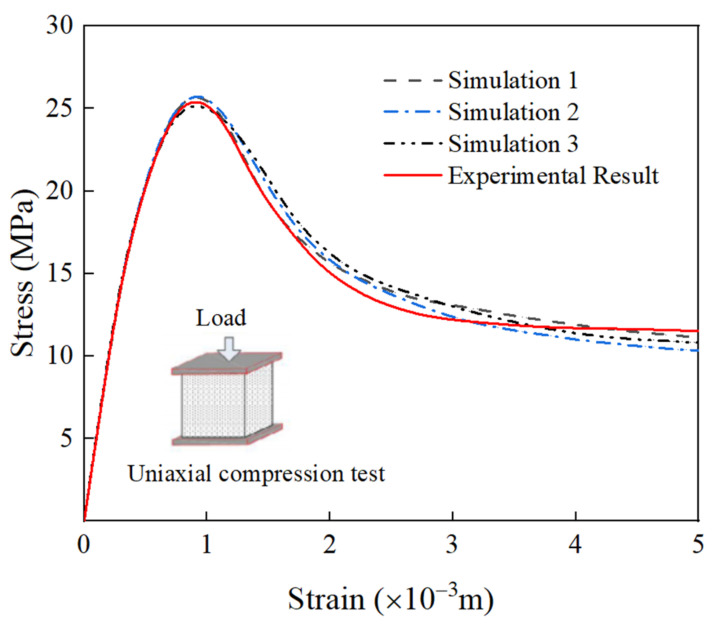
Verification of the meso-model of RAC.

**Figure 7 materials-15-05710-f007:**
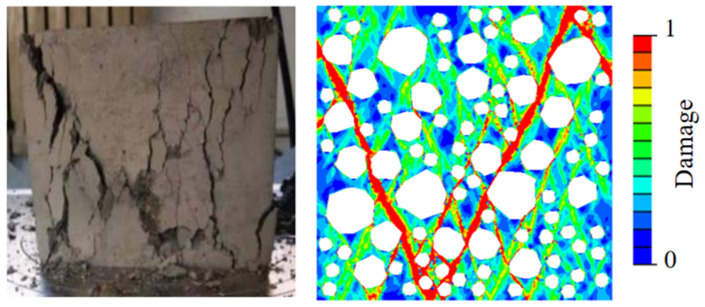
A comparison of the failure modes of compressive specimens.

**Figure 8 materials-15-05710-f008:**
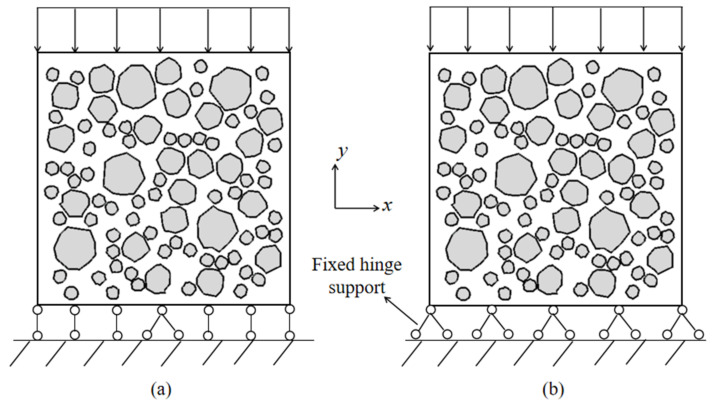
Boundary conditions and loading mode of RAC. (**a**) Loading method 1; (**b**) Loading method 2.

**Figure 9 materials-15-05710-f009:**
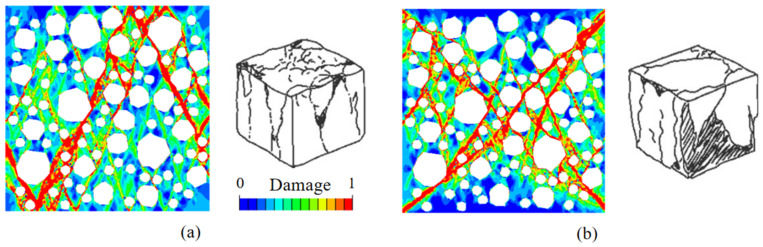
Two representative failure modes of uniaxial compression under different boundary conditions. (**a**) Failure mode 1; (**b**) Failure mode 2.

**Figure 10 materials-15-05710-f010:**
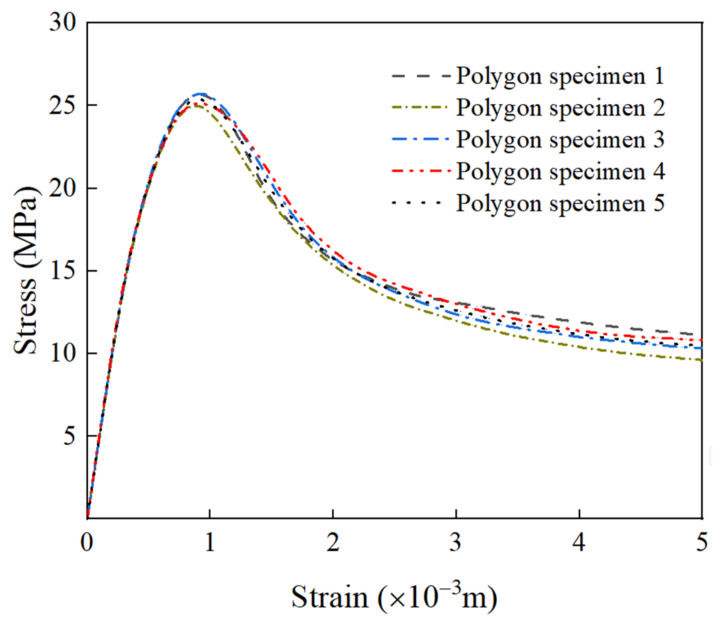
The stress–strain curve of random distribution of RCA.

**Figure 11 materials-15-05710-f011:**
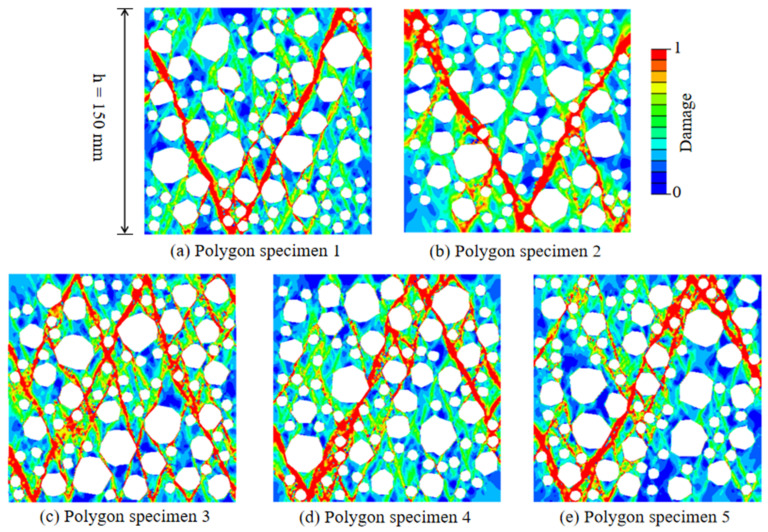
The failure patterns of RAC under different random distribution of RCA.

**Figure 12 materials-15-05710-f012:**
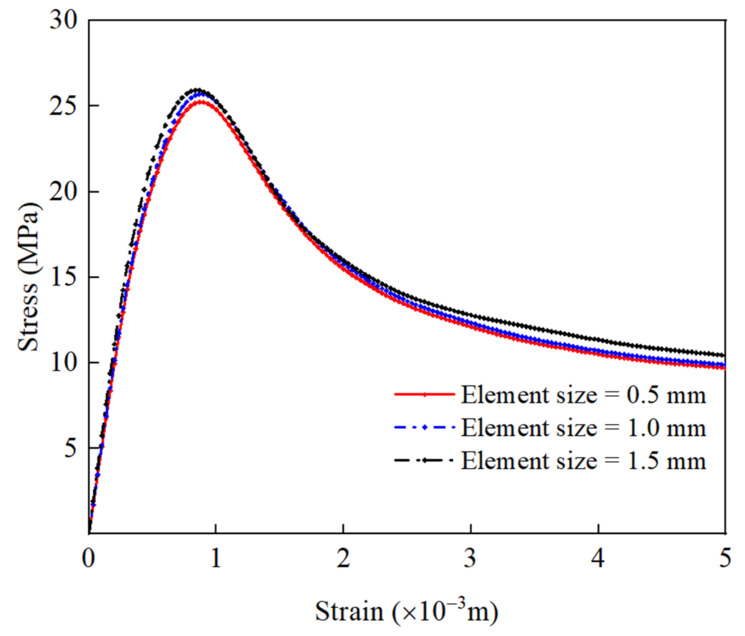
The effect of the mesh size on the stress–strain curve of RAC under uniaxial compression.

**Figure 13 materials-15-05710-f013:**
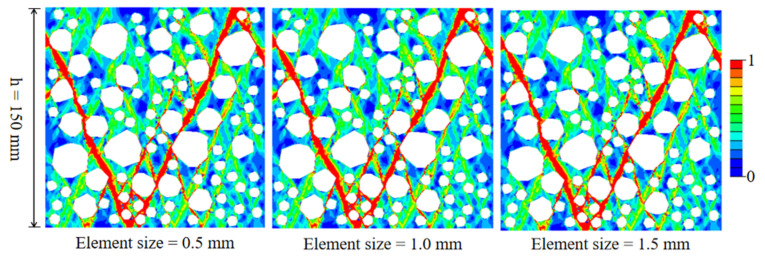
The effect of the mesh size on the failure mode of RAC under uniaxial compression.

**Figure 14 materials-15-05710-f014:**
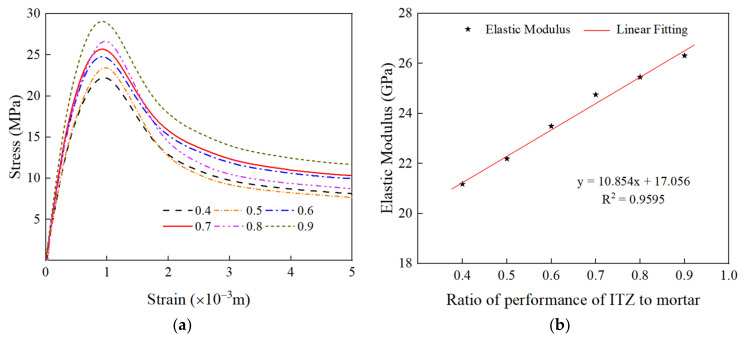
The effect of the strength of ITZ on the stress–strain curve of RAC under uniaxial compression. (**a**) Stress–strain curve; (**b**) linear fitting of elastic modulus; (**c**) linear fitting of peak compressive strength.

**Figure 15 materials-15-05710-f015:**
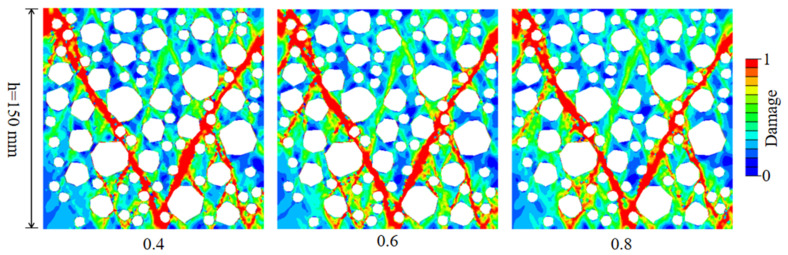
The effect of the strength of ITZ on failure mode of RAC under uniaxial compression.

**Figure 16 materials-15-05710-f016:**
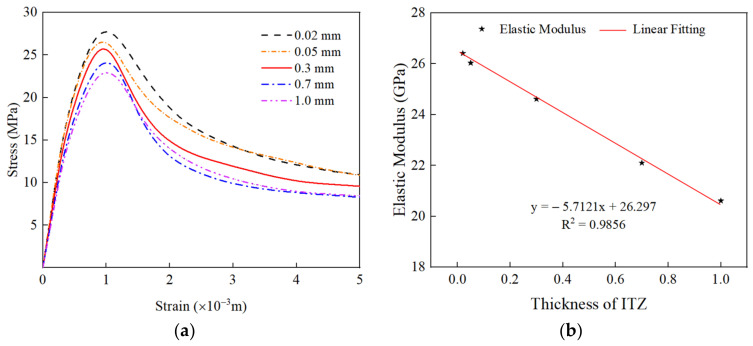
The effect of the thickness of ITZ on the stress–strain curve of RAC under uniaxial compression. (**a**) Stress–strain curve; (**b**) linear fitting of the elastic modulus; (**c**) linear fitting of the peak compressive strength.

**Figure 17 materials-15-05710-f017:**
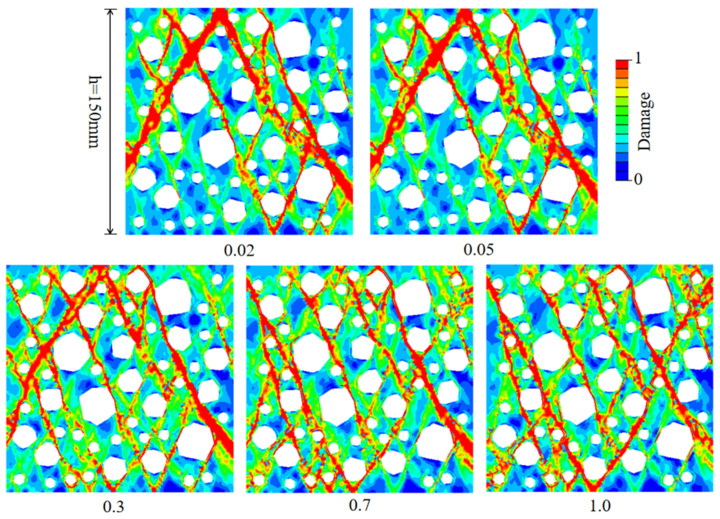
The effect of ITZ thickness on the failure mode of RAC under uniaxial compression.

**Figure 18 materials-15-05710-f018:**
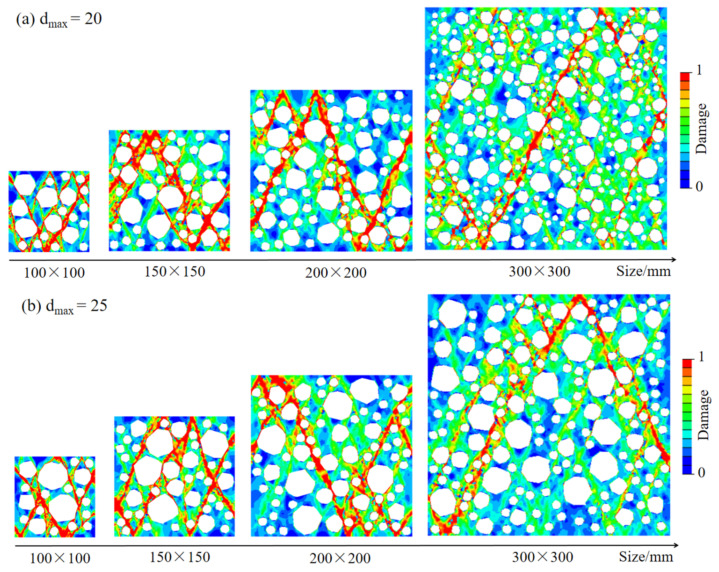
The compression failure modes of RAC under different sizes.

**Figure 19 materials-15-05710-f019:**
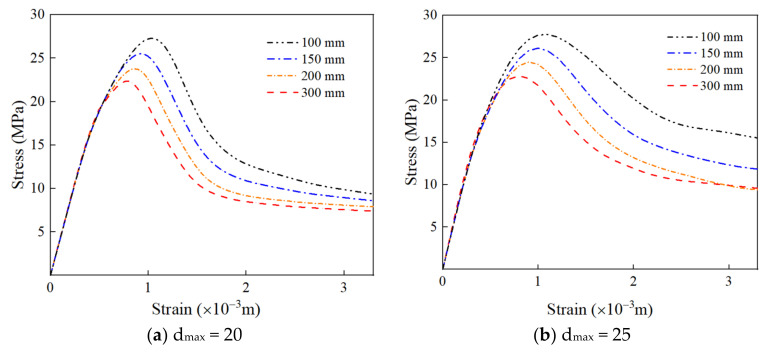
The stress–strain curve of RAC under different sizes.

**Figure 20 materials-15-05710-f020:**
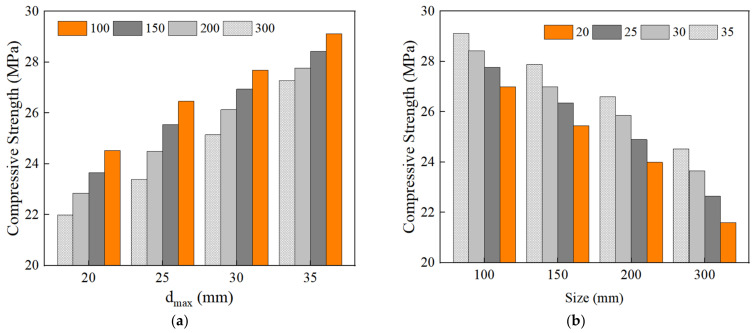
The effect of the maximum aggregate size on the compressive strength of RAC. (**a**) The effect of the maximum aggregate size on the compressive strength of RAC; (**b**) the effect of the specimen size on the compressive strength of RAC.

**Figure 21 materials-15-05710-f021:**
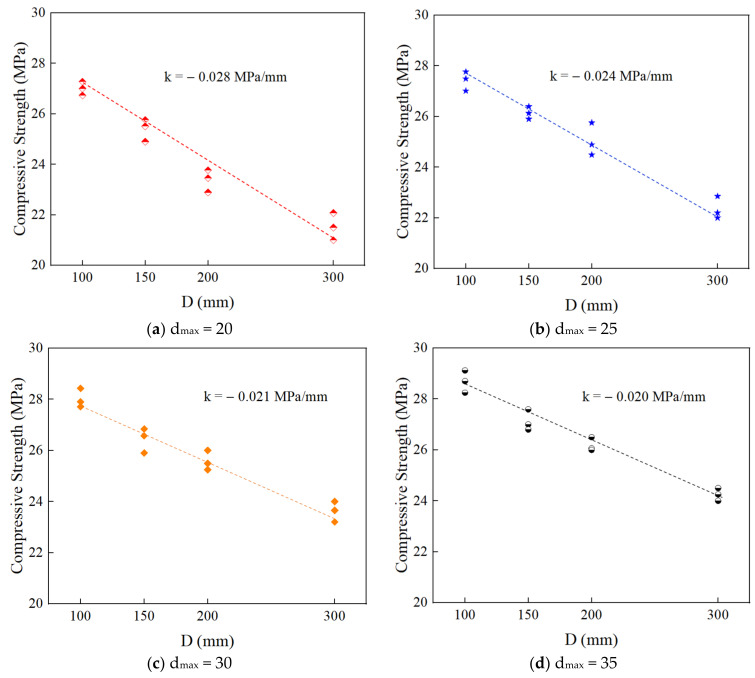
The relationship between the compressive strength and the specimen size of RAC.

**Figure 22 materials-15-05710-f022:**
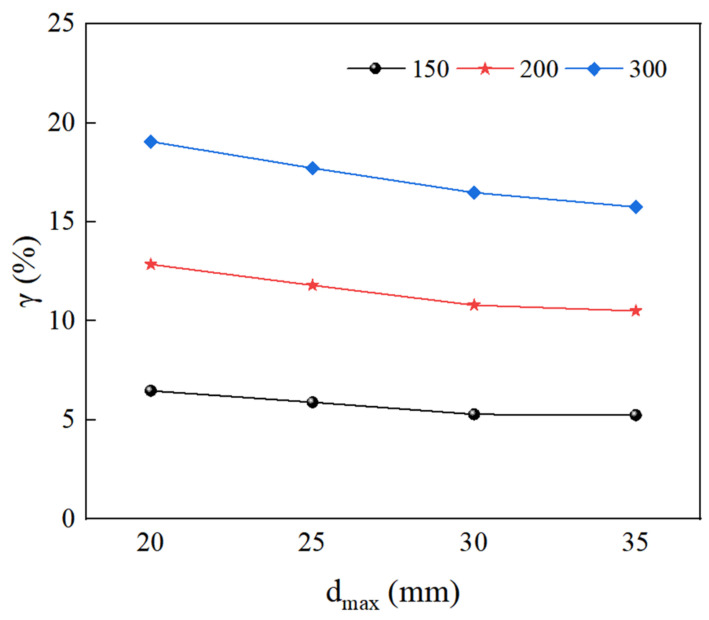
The size effect degree of RAC under different maximum aggregate sizes.

**Figure 23 materials-15-05710-f023:**
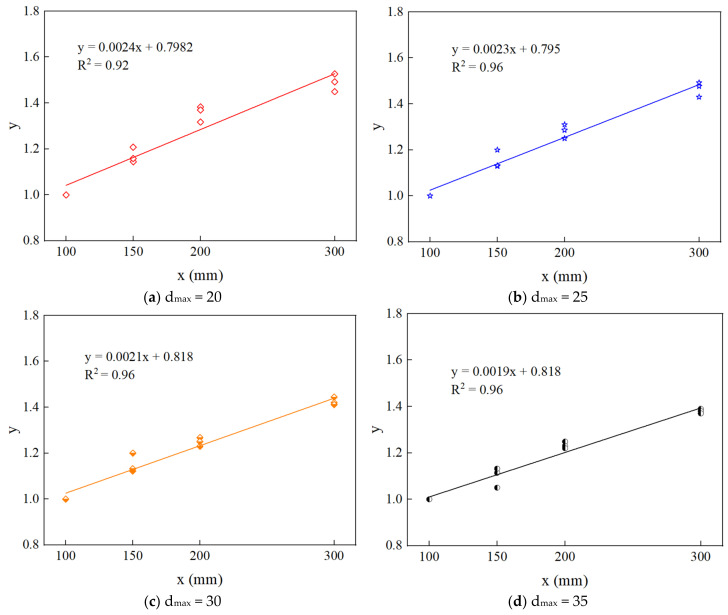
Parameter fitting equation of the Bažant size effect law.

**Figure 24 materials-15-05710-f024:**
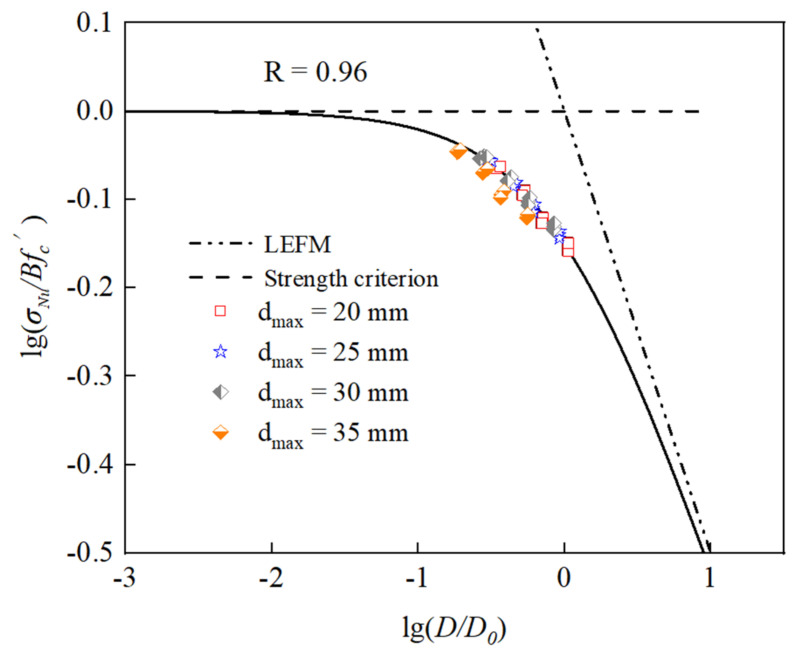
Verification of Bažant’s size effect law for RAC.

**Table 1 materials-15-05710-t001:** Mixture prepositions of RAC.

Group	Mixture Proportion (kg/m^3^)				
	Water	Cement	Sand	RCA	Water Reducing Agent
RAC	210	420	639	1136	4.2

**Table 2 materials-15-05710-t002:** Twenty-eight-day compressive strength of RAC and mortar.

Group	Compressive Strength (MPa)		σ	CV (%)
	Test Value	Average Value		
RAC	26.41	25.9	0.56	2.16
	25.12			
	26.17			
Mortar	28.42	29.0	0.62	2.14
	29.87			
	28.71			

**Table 3 materials-15-05710-t003:** The strength proportional relationship between components of RAC.

Materials	Mortar	Adhered Mortar	New-ITZ	Old-ITZ
Proportional strength	1	1.1	0.7	0.75

**Table 4 materials-15-05710-t004:** Meso-material parameters of RAC.

Materials	Elastic Modulus GPa	Poisson’s Ratio	Compressive Strength MPa	Tensile Strength MPa
NCA	70	0.16	70	7.0
Mortar	20.0	0.22	29	2.9
Adhered mortar	22.5	0.22	32	3.2
New-ITZ	14.0	0.2	19	1.9
Old-ITZ	15.0	0.2	22	2.2

**Table 5 materials-15-05710-t005:** Meso-mesh statistics of RAC.

Size of Mesh/mm	Number of Mesh	Number of Cell Nodes	Computation Time/min
0.5	118,916	127,835	240
1	35,313	40,609	100
1.5	18,446	23,426	30

**Table 6 materials-15-05710-t006:** The statistics of peak stress and peak strain of RAC with different ITZ thickness.

Thickness of ITZ mm	Peak Stress MPa	Peak Strain	Residual Stress MPa	Elastic Modulus GPa
0.02	27.71	0.001	10.13	26.41
0.05	26.51	0.00095	10.11	26.03
0.3	25.70	0.00095	8.90	24.41
0.7	24.07	0.001	7.78	21.90
1.0	22.92	0.001	8.05	20.91

**Table 7 materials-15-05710-t007:** Theoretical parameters of the Bažant size effect law.

d_max_/mm	A	B	C	D
20	0.00264	0.7513	1.15	285
25	0.00239	0.7741	1.14	324
30	0.00223	0.79092	1.12	355
35	0.00205	0.80833	1.11	543

## Data Availability

All the data are available within the manuscript.

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
