# Peer review of "Uniaxial Compression Failure and Size Effect of Recycled Aggregate Concrete Based on Meso-Simulation Analysis"

_materials, 2022, doi:10.3390/ma15165710_

Round 1

Reviewer 1 Report

Authors have carried out a comprehensive and in depth research analysis, however the following observations were made

1)      In the abstract, the opening remarks should be a brief background and research problem

2)      In the first line of the introduction, please define RAC firstly

3)      The review of previous studies can be enhanced, more literature should be added which should be latest

4)      The research problem, research gaps and research objectives should be rewritten

5)      How are the models/analysis validated?

6)      Please explain the practical applications of your research under a sub-section and include in abstract and conclusions

Author Response

General comments: Authors have carried out a comprehensive and in depth research analysis, however the following observations were made.

Response: Thank you for your valuable suggestions to improve the quality of the manuscript. We have taken time to think through all of your comments and carefully revised the manuscript as you suggested. Specific revisions are as follows.

Point 1:  In the abstract, the opening remarks should be a brief background and research problem.

Response 1: Thank you for your comments. According to your suggestion, we have added a brief background and research problem in the opening remarks of the abstract, as follows. “Recycled aggregate concrete (RAC) is a kind of five-phase composite material at the meso-level. It has more complex interfacial transition zone (ITZ) than ordinary aggregate concrete (NAC), which is an important factor affecting the meso-failure of RAC. In addition, the aggregate size plays an important role in the nonlinear mechanical behavior of concrete, which is closely related to the size effect”.

Point 2:  In the first line of the introduction, please define RAC firstly.

Response 2: Thank you for your comments. According to your suggestion, we have defined RAC in the first line of the introduction. The bibliographic citations of Wang (2019) and Fang (2018) have been added.

Point 3:  The review of previous studies can be enhanced, more literature should be added which should be latest.

Response 3: Thank you for your comments. According to your suggestion, we have added the first paragraph in the introduction section to review the importance of RCA and related research, and to analyze the main differences between RCA and natural aggregate. we have added the latest literature reviews of Ma (2022), Bu (2022), Piccinali (2022), Tran (2022), Ahmed (2022), and Xu (2022) in the introduction section.

Point 4:  The research problem, research gaps and research objectives should be rewritten.

Response 4: Thank you for your comments. According to your suggestion, we have rewritten the last paragraph of the introduction, which makes the research problems, research gaps and research objectives of this paper more prominent. 

Point 5:  How are the models/analysis validated?

Response 5: Thank you for your comments. The verification of the meso-numerical model established by the manuscript is mainly based on the following ideas. 1) the peak strength of RAC and mortar is tested and the stress-strain curve is obtained by field test; 2) in mesoscopic analysis, when assigning material properties to mesoscopic components, the compressive strength of new mortar is obtained from test 1), while the new and old ITZ, attached mortar, natural aggregate and other components are assigned properties according to existing literature; 3) set appropriate boundary conditions for mesoscopic simulation, and compare the failure patterns of the simulation with those of the real test; 4) compare the simulated stress-strain curve with the strain-strain curve obtained from test 1). In the process of simulation, the material parameters of each component of RAC can be adjusted on the basis of the existing research. In this study, by adjusting the thickness of ITZ to 0.3mm and the strength of new ITZ to 0.7 times that of mortar, the compressive failure morphology and stress-strain curve obtained by meso-simulation are in good agreement with the real test, which confirms the feasibility of this study.

Point 6:  Please explain the practical applications of your research under a sub-section and include in abstract and conclusions.

Response 6: Thank you for your comments. According to your suggestion, we have added section 5.3 to the manuscript to explain the application of this study in practical engineering, and supplemented it in the summary and conclusion.

In summary, we’d also like to thank you for your careful reviewing of our original version and proposing a lot of useful comments, which are very beneficial to improve the quality of our paper.

Reviewer 2 Report

The manuscript entitled "Uniaxial compression failure and size effect of recycled aggregate concrete based on meso-simulation analysis" presents an experimental study conducted on the modeling of concrete with recycled aggregates. However, the introduction section doesn’t include a clear overview of the previous studies related to the use of recycled aggregates in concrete, and many other issues must be addressed. The paper needs major revisions before it is processed further, some comments follow:

Abstract: The abstract is written qualitatively. The majority of the qualitative statements should be modified for quantified result comparisons. Please add some quantitative comparisons related to the following sentences: "tends to slow down", "with the increase of the maximum aggregate size, the nominal compressive strength of RAC increases".

Introduction Section

The introduction section includes an exhaustive overview of the effect of aggregates in concrete and the previous studies on this topic. However, the authors should introduce a paragraph related to the importance of aggregates recycling and the previous studies related to this topic. (As stated in the title, the study is focussing on recycled aggregates, but this topic wasn’t addressed in the introduction). Which are the main differences between virgin and recycled aggregates? Which characteristics of recycled aggregates have been addressed in previous studies? etc. These questions should be addressed in the introduction of this study. Please consider the following studies: DOI: 10.3390/ma15113929, eBook ISBN: 9780128241066.

2. Establishment and verification of meso-model of RAC Section

Line 232: "the sand ratio was 36%," please state which type of percentage is presented as weight or volume. Please replace the term ratio with the amount or another suitable term.

"particle size of RCA was 5-20mm" – please provide the particle size distribution.

Lines 234-235: "peak stress of RAC is 25.9MPa" please provide deviation values, also for mortar value.

Lines 237-240: The sentence is unclear and hard to follow, please replace it with a table. Show clear value and comparisons in a more schematic and clear presentation.

Figures 6 to 9 – what is the meaning of the color bar? Please provide a clear description of the differences between 0 and 1.

Conclusions

The conclusion section can be improved. Please provide a quantitative evaluation of the results of the studies. Please consider the suggestions for the abstract.

 ·       Please provide some future directions and limitations of the study.

Author Response

General comments: The manuscript entitled "Uniaxial compression failure and size effect of recycled aggregate concrete based on meso-simulation analysis" presents an experimental study conducted on the modeling of concrete with recycled aggregates. However, the introduction section doesn’t include a clear overview of the previous studies related to the use of recycled aggregates in concrete, and many other issues must be addressed. The paper needs major revisions before it is processed further, some comments follow.

Response: Thank you for your valuable suggestions to improve the quality of the manuscript. We have taken time to think through all of your comments and carefully revised the manuscript as you suggested. Specific revisions are as follows.

Abstract

Point 1:  The abstract is written qualitatively. The majority of the qualitative statements should be modified for quantified result comparisons. Please add some quantitative comparisons related to the following sentences: "tends to slow down", "with the increase of the maximum aggregate size, the nominal compressive strength of RAC increases".

Response 1: Thank you for your comments. According to your suggestion, we have checked the abstract and added some quantitative comparisons to the corresponding sentences.

Introduction Section

Point 2:  The introduction section includes an exhaustive overview of the effect of aggregates in concrete and the previous studies on this topic. However, the authors should introduce a paragraph related to the importance of aggregates recycling and the previous studies related to this topic. (As stated in the title, the study is focussing on recycled aggregates, but this topic wasn’t addressed in the introduction). Which are the main differences between virgin and recycled aggregates? Which characteristics of recycled aggregates have been addressed in previous studies? etc. These questions should be addressed in the introduction of this study. Please consider the following studies: DOI: 10.3390/ma15113929, eBook ISBN: 9780128241066.

Response 2: Thank you for your comments. According to your suggestion, we have added the first paragraph in the introduction section to review the importance of RCA and related research, and to analyze the main differences between RCA and natural aggregate. The past studies of Grabiec (2012), Li (2016), Dimitriou (2018), Kim (2018), Ma (2022), Bu (2022), Piccinali (2022), Tran (2022), Ahmed (2022), and Xu (2022) have been added to support the introduction section.

Point 3: Line 232: "the sand ratio was 36%," please state which type of percentage is presented as weight or volume. Please replace the term ratio with the amount or another suitable term.

Response 3: Thank you for your comments. The sand ratio described in this paper refers to the weight percentage of fine aggregate to total aggregate. According to your suggestion, we have supplemented the proportion of mixture of RAC in this paper, as shown in Table 1, and revised the description of the sand ratio in this paper.

Point 4: "particle size of RCA was 5-20mm" – please provide the particle size distribution.

Response 4: Thank you for your comments. According to your suggestion, we have provided the aggregate gradation curve in the section of “2.3 Verification of meso-model” and explained the granulometric properties of RCA in details, as shown in Figure 5.

Point 5: Lines 234-235: "peak stress of RAC is 25.9MPa" please provide deviation values, also for mortar value.

Response 5: Thank you for your comments. According to your suggestion, we have provided the test values, deviation values, and coefficient of variation of each specimen in Table 2.

Point 6: Lines 237-240: The sentence is unclear and hard to follow, please replace it with a table. Show clear value and comparisons in a more schematic and clear presentation.

Response 6: Thank you for your comments. According to your suggestion, we have added Table 3 to reflect the ratio between new and old ITZ and new and old mortar.

Point 7: Figures 6 to 9 – what is the meaning of the color bar? Please provide a clear description of the differences between 0 and 1.

Response 7: Thank you for your comments. We are very sorry that we did not express the meaning of the color bar clearly. When meso-numerical simulation is carried out by ABAQUS, the damage variable corresponding to the color bar can reflect the damage degree of the simulated sample. When the damage variable is 0, it means that the sample has not been damaged, and when the damage variable is 1, it means that the sample is completely destroyed. According to your suggestion, we have added made a supplementary explanation in section 2.3.

Conclusions

Point 8: The conclusion section can be improved. Please provide a quantitative evaluation of the results of the studies. Please consider the suggestions for the abstract. 

Response 8: Thank you for your comments. According to your suggestion, we have considered the suggestions for the abstract and improved the conclusion section through quantitative evaluation.

Point 9: Please provide some future directions and limitations of the study. 

Response 9: Thank you for your comments. According to your suggestion, we have added the limitations and some future research directions of the study at the end of the manuscript.

In summary, we’d also like to thank you for your careful reviewing of our original version and proposing a lot of useful comments, which are very beneficial to improve the quality of our paper.

Round 2

Reviewer 1 Report

Authors have addressed all comments raised by me

Author Response

We’d like to thank you for your careful reviewing of our original version and proposing a lot of useful comments, which are very beneficial to improve the quality of our paper. Thank you very much.

Reviewer 2 Report

The authors consider most of my comments and the article was improved accordingly. However, the label related to the color bar (damage) should be introduced to all figures that have this axis. Moreover, some comments should be introduced in the text related to the correspondence between 0 and 1 and the damage to the sample (for example at 0 - no damage, at 1 - 100% damage, the integrity of the matrix is destroyed...).

Best,

Author Response

General comments: The authors consider most of my comments and the article was improved accordingly. However, the label related to the color bar (damage) should be introduced to all figures that have this axis. Moreover, some comments should be introduced in the text related to the correspondence between 0 and 1 and the damage to the sample (for example at 0 - no damage, at 1 - 100% damage, the integrity of the matrix is destroyed...).

Response: Thank you for your valuable suggestions to improve the quality of the manuscript. We have taken time to think through all of your comments and carefully revised the manuscript as you suggested. Specific revisions are as follows.

Point 1: The label related to the color bar (damage) should be introduced to all figures that have this axis. 

Response 1: Thank you for your comments. According to your suggestion, we have checked all the figures with color bar in the manuscript, and added the label (Damage) to the color bar in Figure 7, 9, 11, 13, 15, and 17. 

Point 2: Some comments should be introduced in the text related to the correspondence between 0 and 1 and the damage to the sample (for example at 0 - no damage, at 1 - 100% damage, the integrity of the matrix is destroyed...). 

Response 2: Thank you for your comments. According to your suggestion, we have supplemented the damage degree and damage state corresponding to damage factors 0, 0-1, and 1 in Section 2.3. In addition, the bibliographic citations of Benkemoun (2016), Chen (2018), Li (2018), Jin (2019), Jin (2020), and Yu (2021) have been added to to support the statement in the revised manuscript.

In summary, we’d also like to thank you for your careful reviewing of our original version and proposing a lot of useful comments, which are very beneficial to improve the quality of our paper.